# Proteomics biomarker discovery for individualized prevention of familial pancreatic cancer using statistical learning

Chung Shing Rex Ha[1,2,3☯]*, Martina Müller-Nurasyid[1,2,4,5☯], Agnese Petrera[6], Stefanie M. Hauck[6], Federico Marini[1,7], Detlef K. Bartsch[8], Emily P. Slater[8‡], Konstantin Strauch[1,2,3‡]

**1** Institute of Medical Biostatistics, Epidemiology and Informatics (IMBEI), University Medical Center, Johannes Gutenberg University, Mainz, Germany, **2** Institute of Genetic Epidemiology, Helmholtz Zentrum München—German Research Center for Environmental Health, Neuherberg, Germany, **3** Faculty of Medicine, Institute for Medical Information Processing, Chair of Genetic Epidemiology, Biometry, and Epidemiology (IBE), LMU Munich, Munich, Germany, **4** Faculty of Medicine, Institute for Medical Information Processing, Biometry, and Epidemiology (IBE), LMU Munich, Munich, Germanys, **5** Faculty of Medicine, Institute for Medical Information Processing, Pettenkofer School of Public Health Munich, Biometry, and Epidemiology (IBE), LMU Munich, Munich, Germany, **6** Research Unit Protein Science and Metabolomics and Proteomics Core Facility, Helmholtz Zentrum München—German Research Center for Environmental Health, Neuherberg, Germany, **7** Research Center for Immunotherapy (FZI), University Medical Center, Johannes Gutenberg University, Mainz, Germany, **8** Department of Visceral-, Thoracic- and Vascular Surgery, Philipps University, Marburg, Germany

☯ These authors contributed equally to this work.
‡ These authors also contributed equally to this work.
* rexha@uni-mainz.de

**Data Availability Statement:** The individual-level data of the FaPaCa study are not publicly available, because the data contain sensitive patient data which underlie data protection rules. This is in

## Abstract

### Background

The low five-year survival rate of pancreatic ductal adenocarcinoma (PDAC) and the low diagnostic rate of early-stage PDAC via imaging highlight the need to discover novel biomarkers and improve the current screening procedures for early diagnosis. Familial pancreatic cancer (FPC) describes the cases of PDAC that are present in two or more individuals within a circle of first-degree relatives. Using innovative high-throughput proteomics, we were able to quantify the protein profiles of individuals at risk from FPC families in different potential pre-cancer stages. However, the high-dimensional proteomics data structure challenges the use of traditional statistical analysis tools. Hence, we applied advanced statistical learning methods to enhance the analysis and improve the results' interpretability.

### Methods

We applied model-based gradient boosting and adaptive lasso to deal with the small, unbalanced study design via simultaneous variable selection and model fitting. In addition, we used stability selection to identify a stable subset of selected biomarkers and, as a result, obtain even more interpretable results. In each step, we compared the performance of the different analytical pipelines and validated our approaches via simulation scenarios.

accordance with the local ethic vote and the regulations of the FaPaCa registry. Patients' characteristics are available upon request from the FaPaCa study registry (contact information: fapaca@med.uni-marburg.de). The aggregated results from the FaPaCa study and the full codes for the simulation are available in its Supporting Information files.

**Funding:** This research was supported by a grant from the Wilhelm Sander-Stiftung to EPS, DKB, and KS (No. 2018.022.1) and a generous donation from the GAUFF-Foundation. Furthermore, this research was supported within the Munich Center of Health Sciences (MC-Health), Ludwig Maximilian University of Munich, as part of LMUinnovativ, as well as by a grant from the German Research Foundation to FM (No. 318346496, SFB1292/2 TP19N). The funders had no role in study design, data collection and analysis, decision to publish, or preparation of the manuscript.

**Competing interests:** The authors have declared that no competing interests exist.

## Results

In the simulation study, model-based gradient boosting showed a more accurate prediction performance in the small, unbalanced, and high-dimensional datasets than adaptive lasso and could identify more relevant variables. Furthermore, using model-based gradient boosting, we discovered a subset of promising serum biomarkers that may potentially improve the current screening procedure of FPC.

## Conclusion

Advanced statistical learning methods helped us overcome the shortcomings of an unbalanced study design in a valuable clinical dataset. The discovered serum biomarkers provide us with a clear direction for further investigations and more precise clinical hypotheses regarding the development of FPC and optimal strategies for its early detection.

## Background

Pancreatic ductal adenocarcinoma (PDAC) is a challenging tumor entity with an increasing incidence and a dismal prognosis [1,2]. The overall five-year survival rate of PDAC patients is less than 5%, which can be attributed to late clinical symptoms, low resection rates, and poor response to radio- and chemotherapy [3]. One of the greatest risk factors for developing PDAC is a positive family history [4–7]. When two or more first-degree relatives that do not fulfil the criteria for another inherited tumor syndrome have PDAC, this is called familial pancreatic cancer (FPC) [7,8].

The German National Case Collection of Familial Pancreatic Cancer (FaPaCa), a tumor registry, was established to investigate the phenotype and genotype of FPC families [6,9]. Research has focused on the underlying gene defects, biomarker development, and the evaluation of prospective PDAC screening programs for members of such families [5]. Current whole genome and whole exome sequencing data suggest that FPC is genetically highly heterogeneous, with no single predisposing gene responsible for the occurrence of the disease in families [10,11].

Early-stage PDAC and its high-grade precursor lesions are asymptomatic and difficult to diagnose with current imaging techniques [12]. However, the quality of diagnosis has a direct influence on the decision of whether or not to perform pancreatic resection surgery [7]. A late diagnosis may increase the risk of complications during the operation, while a misdiagnosis, e.g., in the case of pancreatitis, may lead to an unnecessary resection of the pancreas and thus impaired quality of life for patients [1,7]. Consequently, clinical experts are still looking for effective biomarkers to improve the accuracy of the diagnosis and inform the decision to intervene surgically.

Our previous studies found two potential serum biomarkers, TIMP1 and LCN2, whose inclusion improve the current diagnostic tools in established screening procedures for early-stage PDAC [13,14]. The successful discovery of novel protein biomarkers motivated us to apply high-throughput proteomics in our study. This innovative technique allows us to obtain the comprehensive protein profile of each individual and perform analysis on a large number of proteins. However, the high-dimensional data poses a challenge for statistical analysis and is not suited for standard measurement in diagnosis or treatment assessment. Nevertheless, it is reasonable to assume that even a subset of proteins selected with well-guided statistical

techniques from the protein profile could help improve the predictive capability of screening procedures in FPC due to the strong pathophysiological interplay of biomarkers.

Since PDAC is often diagnosed at an advanced stage when patients show symptoms of major changes in metabolic processes, e.g. cachexia, proteomic and/or metabolic profiles of PDAC patients may already have been altered substantially [15–17]. Therefore, we aim to discover a robust subset of serum biomarkers for detection of significant lesions prior to or at an early, asymptomatic stage of cancer. To achieve this goal, we investigated individuals at risk (IARs) of FPC families with three different phenotypes: those without or with lesions detected on imaging by either magnetic resonance imaging (MRI) and/or endosonography, and those with histologically significant lesions. The latter include high-grade pancreatic intraepithelial neoplasms (PanIN) and intraductal papillary-mucinous neoplasms (IPMN) with dysplasia which are considered true precursor lesions of PDAC [12,18–21]. Significant lesions, however, are rarely discovered in IARs, as they can only be confirmed histologically after the pancreatic surgery. This fact leads to small sample sizes and an unbalanced study design in the FaPaCa data.

Small sample sizes are a common challenge in high-throughput 'omics studies and lead to the high-dimensional data ($p>n$) problem [22,23]. At the same time, an unbalanced study design reduces the power of statistical tests [24]. Therefore, classical statistical tools such as hypothesis testing or generalized linear regression models are not suited for high-dimensional data with small and unbalanced sample sizes. Instead, regularized regression models, such as ridge and lasso regression, are a popular tool to tackle the small sample size problem via the introduction of penalty terms [22,25]. A further benefit of regularized regression models is that they can also perform variable selection to identify the most predictive biomarkers while fitting the model parameters [25]. However, despite a robust performance against small sample sizes, the variable selection procedure becomes unstable under an unbalanced study design. This problem becomes more severe in high-throughput 'omics studies, where biomarkers are usually highly intercorrelated.

Besides regularized regression models, model-based gradient boosting (mboost) is an alternative approach to overcome the above-mentioned limitations [18]. The iterative learning property allows mboost models to learn from small datasets with unbalanced group sizes [26]. Further, the integration of the boosting algorithm into the generalized additive model (GAM) increases the interpretability of the boosted regression models compared to the conventional boosted tree model [27].

To discover potential serum biomarkers for FPC, we compared the prediction and variable selection performance between the adaptive lasso and mboost in a simulation study of small and unbalanced samples in conjunction with high-dimensional data, i.e., a large number of variables. Afterwards, we applied the best-performing method to the FaPaCa proteomics data. To control the likelihood of false positive discovery, we additionally applied stability selection to identify the stable subset of the biomarkers.

## Materials and methods

### Materials

**FaPaCa study.** First-degree relatives of an affected patient of a FPC family and members of a FPC family carrying a predisposing mutation such as *BRCA2*, independent of the degree of relationship, were classified as individuals at risk (IARs). IARs older than 18 years were encouraged to participate in a prospective screening program conducted at the Department of Surgery, Philipps University of Marburg. The screening started at age 40 years until 2016, and thereafter either at age 50 years or 10 years before the earliest age of onset of PDAC in the

family, whichever came first [5]. The screening program included an annual physical examination, collection of blood samples, determination of serum HbA1c, amylase, GOT, GPT, bilirubin, and CA19-9, and imaging with MRI plus magnetic resonance cholangiopancreatography and endosonography as described previously [5]. The screening program was restricted to mutation carriers if the underlying gene defect in the family was known. In the case of surgical resection, experienced pathologists analyzed the tissue specimens with special regard to the presence of PDAC, PanIN, IPMN, and atypical flat lesions (AFL). The IARs ($n = 83$) were classified as having one of three possible phenotypes: without lesions (w.o; $n_{w.o} = 26$), lesions detected with imaging (L; $n_L = 49$), and histologically significant lesions (HisSig; $n_{HisSig} = 8$).

The FaPaCa registry, including the genetic analyses and the screening program, was approved by the Ethics Committee of the Philipps University of Marburg (No. 36/1997, last amendment 9/2010), and all participants provided written informed consent.

**Proteomics analysis within FaPaCa.** We used a high-throughput proteomics dataset of patients' serum samples in the FaPaCa study for the real-case application. The concentration of different potential protein biomarkers was measured using proximity extension assay technology performed with Olink Cardiometabolic, Cardiovascular III, Immuno-Oncology, and Oncology II panels (Olink Proteomics, Uppsala, Sweden). Assay concentrations were reported in Normalized Protein eXpression values (NPX), Olink's arbitrary unit in log2 scale. A high NPX value corresponds to a high assay concentration. Due to its relative quantification, NPX is only comparable across samples within the same assay, not across samples from different assays. A minimal proportion of missing values due to deviating results ($< 0.5\%$) in the proteomics dataset were reported, and some measurements under the limit of detection (LOD) were obtained. Observed values under the LOD represent values where the linear relationship between the arbitrary unit NPX and the corresponding true assay concentration may no longer hold true. Please refer to Olink's website (https://www.olink.com/resources-support/) for detailed information.

The following data pre-processing steps took place before statistical analysis:

1. Missing values were assumed missing at random and imputed with missForest, which uses the random forest approach [28].

2. Assays with more than 50% of observations below the LOD were removed as recommended by Olink.

Original measurements from assays with less than 50% of observations below the LOD were kept to preserve the data quality. After quality control of the measured biomarkers, 330 protein assays were used for modelling.

## Simulation study

In order to demonstrate the utility of the proposed methods before applying them to the real-case FaPaCa data, we simulated multiple datasets of a binary classification problem using the following data generating mechanism (available in S2 Appendix), which is based on that of Piironen et al. [29]:

$$
\begin{aligned}
f &\sim N(0, 1) \\
y|f &\sim Bernoulli(invlogit(\tau + \sigma \cdot f)) \\
x_j|f &\sim N(\sqrt{\rho_j}f, 1 - \rho_j), \quad for\ j = 1, \ldots, p_{ref} \\
x_j|f &\sim N(0, 1), \quad for\ j = p_{ref} + 1, \ldots, P
\end{aligned}
\tag{1}
$$

$y$ is the observable outcome generated from an unobserved Gaussian distributed latent variable

$f$. The hyperparameters $\tau$ and $\sigma$ control the ratio between the binary classes and signal-to-noise ratio of $f$, and *invlogit* denotes the inverse logistic function. $x_j$ are the marginally Gaussian distributed observable variables, where only the first $p_{ref}$, i.e. $j = 1,\ldots,p_{ref}$, variables are correlated with $f$, i.e., $cor(x_j, f) = \sqrt{\rho_j}$. The correlation between $x_j$ and $x_k$ is $cor(x_j, x_k) = \sqrt{\rho_j \rho_k}$ for all $j \neq k$ and $j, k \in \{1,\ldots,p_{ref}\}$, and 0 otherwise.

To simulate the small, unbalanced, and high-dimensional data in the FaPaCa study, we set the training sample size for model comparison to $n_{train} = 30$ and the number of variables to $P = 500$. A subset of only $p_{ref} = 10$ relevant variables was defined to represent a theoretical subset of relevant biomarkers. We also set $\tau = -8.5$, maintaining an unbalanced class ratio of 80:20 on average, and $\sigma = 10$. The correlation $\rho_i$ between relevant variables and the influencing latent variable was evenly distributed between 0.05 and 0.95 to mimic the highly intercorrelated structure of relevant biomarkers. To validate the trained model, we generated an extra test dataset of size $n_{test} = 20$ for each realization. We used the data generating mechanism (1) to generate 100 realizations. All variables were standardized prior to model fitting.

In the second phase of the simulation study, we controlled the number of falsely selected variables to test the capability of complementary pair stability selection, a modified version of stability selection proposed by Shah and Samworth which eases the required strong exchangeability assumption of irrelevant variables [30]. We simulated multiple datasets using the same generator and the same configurations as in the original simulation study, adjusting the parameters as follows:

1. $N = \{30, 50, 100\}$

2. $P = \{100, 500, 5000\}$

3. $p_{ref} = \{5, 10, 20\}$

4. $\tau = \{-4.5, -6.5, -8.5\}$

5. $\rho = \{low, mid, high\}$

For each parameter adjustment, we only changed one parameter and kept the other parameters fixed. For $\rho$, {low, mid, high} represents uniform distributions within the intervals [0.05, 0.35], [0.35, 0.65], and [0.65, 0.95], respectively. We generated 100 realizations for each scenario.

## Methods

### Simulation study

In the first simulation study, we compared the prediction performance of adaptive lasso and mboost using small, unbalanced, and high-dimensional data with ridge regression as the benchmarking model [25,27,31,32]. More details about ridge regression, adaptive lasso, and mboost can be found in S1 Appendix. Here, we only considered linear base-learners in the boosting model (glmboost) for model simplicity [33]. Throughout the simulation, we used a negative binomial log-likelihood function as the loss function of the binary classification problem. We compared the test prediction performance of the models via averaged receiver operating characteristic (ROC) curves and their AUCs.

In the second simulation study, we used adaptive lasso and glmboost as underlying variable selection methods of stability selection and performed the complementary pairs stability selection (CPSS) with B = 50 subsampling procedures (for details about CPSS, see S1 Appendix) [30,34,35]. According to equation (S1-7), the stability threshold $\pi_{thr} = 0.504$ was determined by per-family error rate PFER = 4 and the number of selected variables $q = 20$ under the

unimodal distribution assumption. To measure the quality of variable selection, we compared the size of the stable subsets and the relevant variable discovery rate (RVDR) of both methods.

$$RVDR = \frac{number\ of\ relevant\ variables\ in\ the\ stable\ subset}{number\ of\ variables\ in\ the\ stable\ subset} \qquad (2)$$

## FaPaCa study

We divided the dataset into three binary classification scenarios: L against HisSig (L-HisSig), w.o against HisSig (w.o-HisSig), and w.o against L (w.o-L). Aiming at biomarker discovery, we applied glmboost because it performed the best in the simulation study. In addition to glmboost, we introduced gamboost, which included both centered linear and centered smooth P-spline base-learners to investigate the non-linear effects of the biomarkers. As in the simulation study, we used the negative binomial log-likelihood as the loss function of the binary classification problem. The optimal fitting iteration $m_{stop}$ of each model was determined via bootstrapping and the learning rate $v$ was set to 0.1, following the recommendation of Schmid and Hothorn 2008 [36]. To compare the test prediction performance between glmboost and gamboost, we estimated the ROC curves and the corresponding averaged AUCs via the repeated four-fold cross-validation. The variables of the training datasets and test datasets were centered accordingly before being fitted to the models.

Moreover, we implemented CPSS with both glmboost and gamboost to identify the respective subsets of stable biomarkers. To conduct a fair comparison, we set the upper bound of the expected number of false discoveries PFER = 2 and the number of selected variables for each subsample q = 10 for glmboost, and *PFER* = 4 and $q$ = 20 for gamboost, as gamboost contained almost twice as many base-learner choices as glmboost in all scenarios (glmboost: 333 base-learners; gamboost: 663 base-learners). Using equation (S1-7), we obtained the stability threshold $\pi_{thr}$ = 0.55 for both methods.

To identify the biological pathway of the resulted subset of stable biomarkers, we performed an over-representation analysis using the Reactome pathway database [37]. In the analysis, all 330 eligible Olink biomarkers were used as background genes, but only 286 were recognized by Reactome. Fisher's exact test was used to determine the statistical significance of the pathways and the Benjamini-Hochberg procedure was applied to control the false discovery rate. The significance level was set at 5%.

## Statistical software

All statistical analyses were performed in the statistical programming language R (version 3.6.3) [38]. We performed ridge regression and adaptive lasso with package **glmnet** [39], mboost with package **mboost** [33], stability selection with package **stabs** [35], and over-representation analysis with package **clusterProfiler** and **ReactomePA** [40–42].

## Results

### Simulation study

**Prediction performance of different methods.**   Fig 1 depicts the test result summary of the 100 realizations in terms of the average ROC curves with a one standard deviation interval. Compared to ridge regression, both glmboost and adaptive lasso showed a better classification performance on the test datasets via variable selection, as evidenced by the corresponding average areas under curve (AUCs). Both models enhanced the model prediction by regularizing most of the irrelevant variables to zero. It is worth noting that adaptive lasso generally tended to include more variables than glmboost, while glmboost tended to select the relevant variables more often

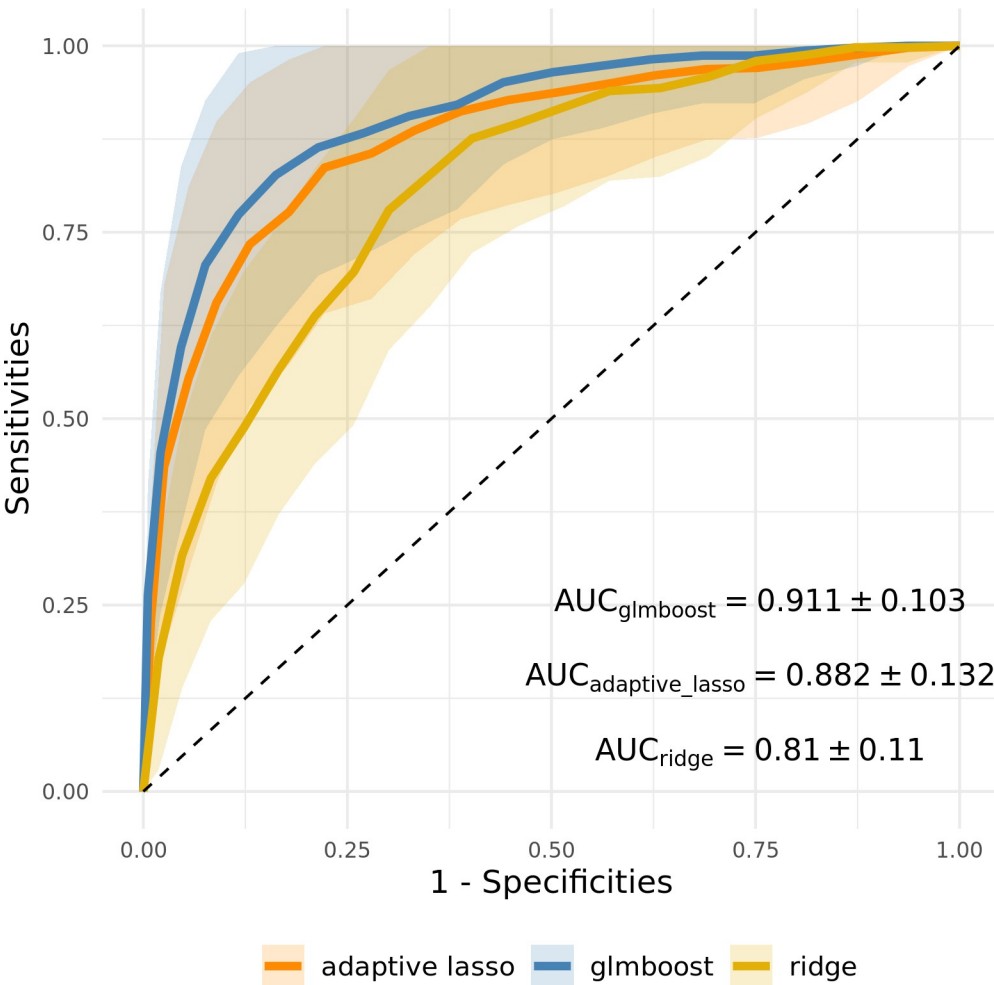

**Fig 1. Averaged ROC curves of test results of the first simulation experiment.** The shaded areas represent the one standard deviation interval of the corresponding methods, estimated via 100 realizations.

compared to adaptive lasso (Fig 2). On average, glmboost achieved a higher selection rate of relevant variables. This possibly explains why glmboost performed better on average than adaptive lasso in this simulation experiment (Fig 1). Nevertheless, the mostly overlapping one standard deviation intervals of both models indicate their similar prediction performance.

## Identifying a stable subset of variables

Despite a massive improvement as a result of using adaptive lasso and glmboost, some variables were still falsely selected, as shown in Fig 2B.

Fig 3 summarizes the results in terms of the number of selected stable variables, i.e., variables with a selection frequency above the stability threshold, in each simulation. Fig 4 summarizes the RVDR in every stable variable subset per simulation. Since it was not guaranteed that a stable variable subset would be identified in every simulation, the presence of empty subsets thus led to an undetermined RVDR in some simulation instances. Therefore, in Fig 4, the color gradient represents the number of simulations with at least one stable variable selected in each scenario. The RVDR concentrated around 1 in most scenarios (Fig 4), indicating that the identified variables in the stable subset are often relevant variables.

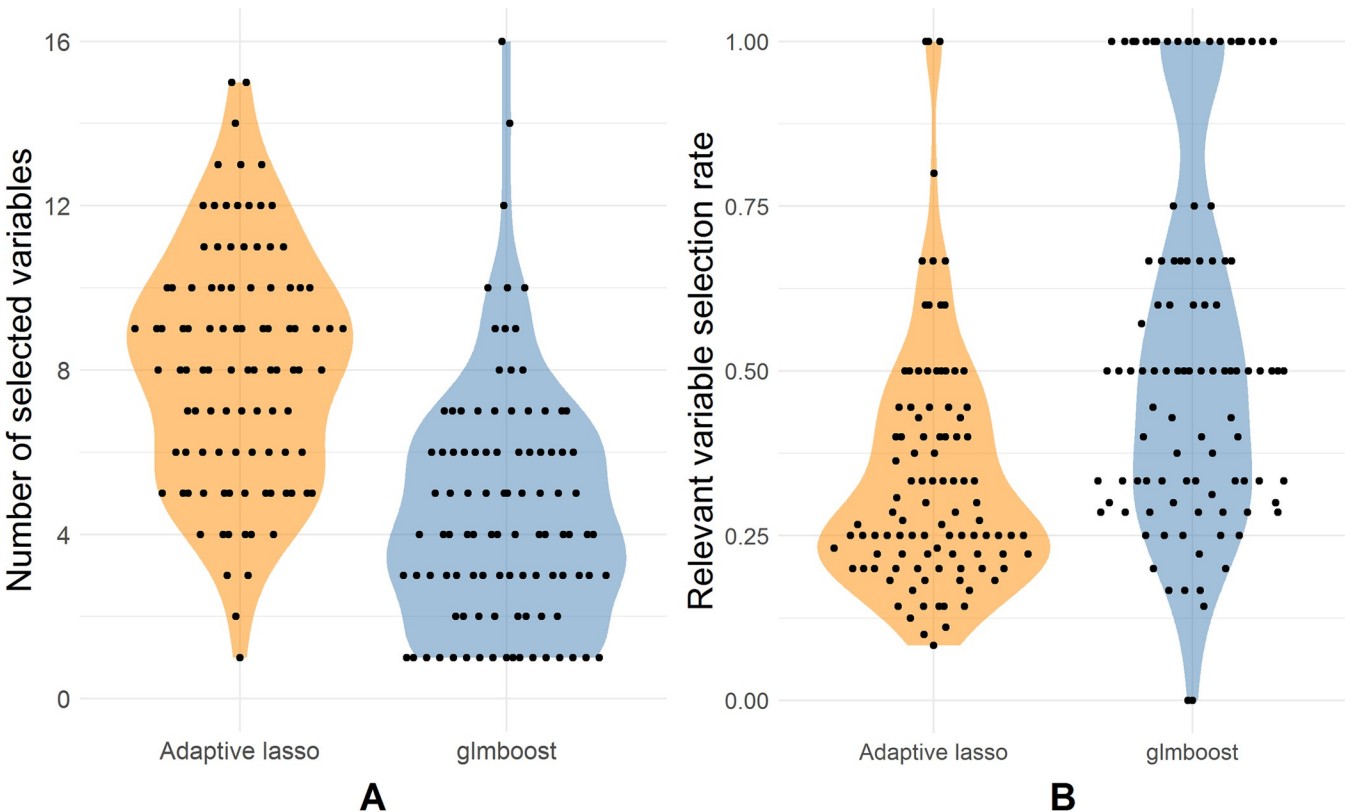

**Fig 2. Summary of numbers of selected variables and selection rates of relevant variables.** (A) Numbers of selected variables per simulation in 100 realizations. (B) selection rates of relevant variables per simulation in 100 realizations.

With regard to the number of selected stable variables (Fig 3), the results of CPSS with adaptive lasso and glmboost were very similar, while glmboost was able to identify more non-empty stable variable subsets. Moreover, glmboost achieved a higher RVDR in most scenarios (Fig 4), while adaptive lasso resulted in rather dichotomous discovery rates. As would be expected, an increase in sample size $N$ improved the identification of stable variable subsets and RVDR for all methods. Still, nearly half of the relevant variables were not identified, even when the sample size $N$ was increased to 100. Also, an improvement in the CPSS's performance could not be achieved by reducing the noise variable (lower $F$). In the ultra-high dimensional case ($P = 5000$), both methods resulted in a large number of empty stable subsets. Surprisingly, the RVDRs decreased when the number of relevant variables $p_{ref}$ increased to 20. To conclude, CPSS worked better with glmboost than with adaptive lasso to identify the truly relevant variables in our simulation study.

## FaPaCa study

Fig 5 shows the averaged ROC curves of both glmboost and gamboost based on the cross-validated prediction results on test datasets in each scenario. Both models had very competitive results according to the overall high average AUCs. gamboost achieved a larger average AUC than glmboost in scenario L-HisSig, while glmboost demonstrated a larger average AUC in scenario w.o-HisSig. Since the estimated one standard deviation intervals of the ROC curves of both models overlapped with each other in every scenario, we could not identify a clear winner between the methods. Both methods could discriminate well and precisely predict lesion status

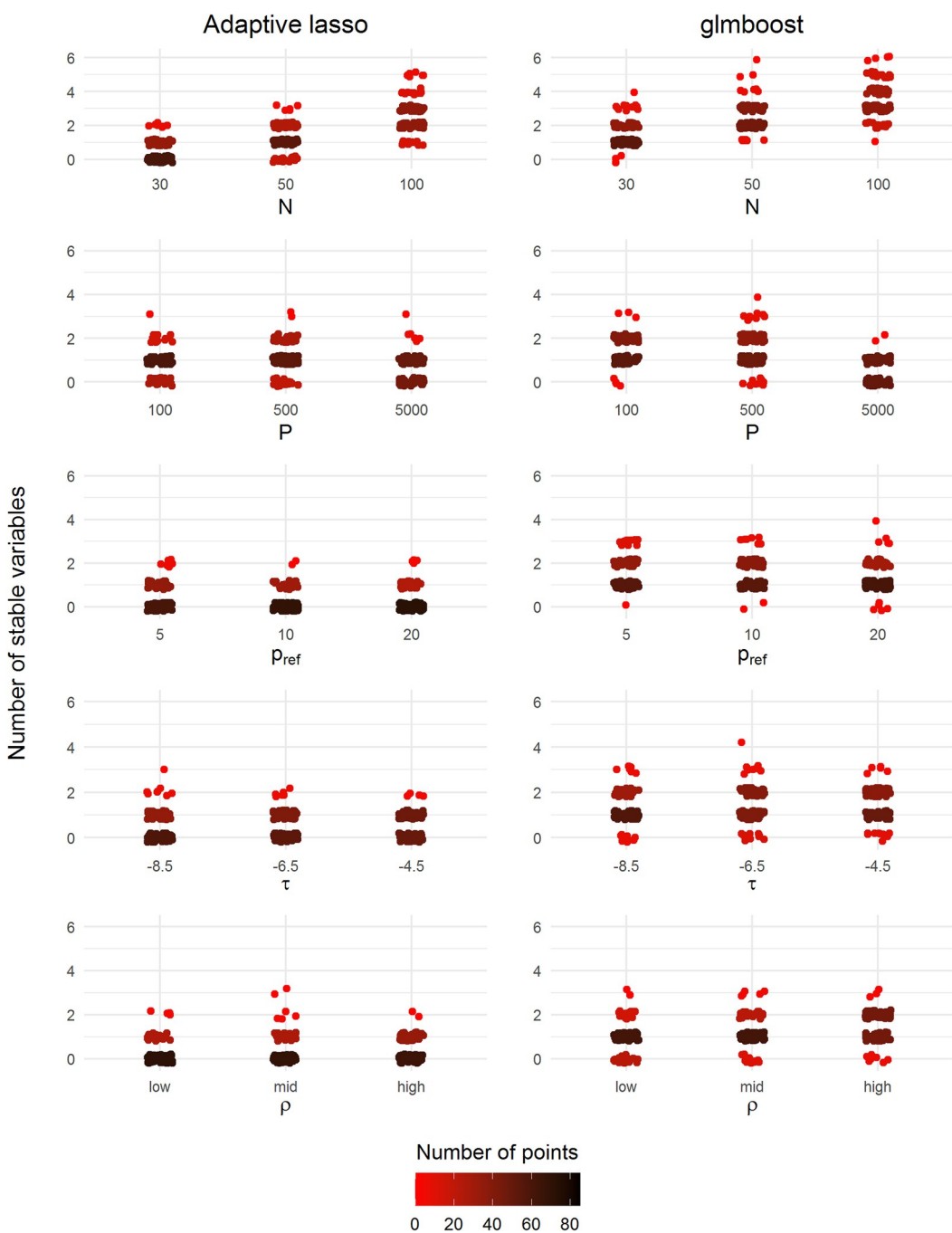

**Fig 3. Number of selected stable variables in all scenarios using adaptive lasso and glmboost.** The colors represent the number of jittered points (red: Low density, black: High density). A total number of 100 realizations of simulated datasets were drawn for each scenario.

in each scenario: the average AUCs of all ROC curves in the scenarios L-HisSig and w.o-HisSig were higher than 0.9. It is also worth mentioning that, while the scenario w.o-L had a larger sample size ($n_{w.o-L} = 75$) and more balanced data structure compared to the other scenarios, the test prediction performance of both models in this scenario was the lowest among all scenarios (Fig 5).

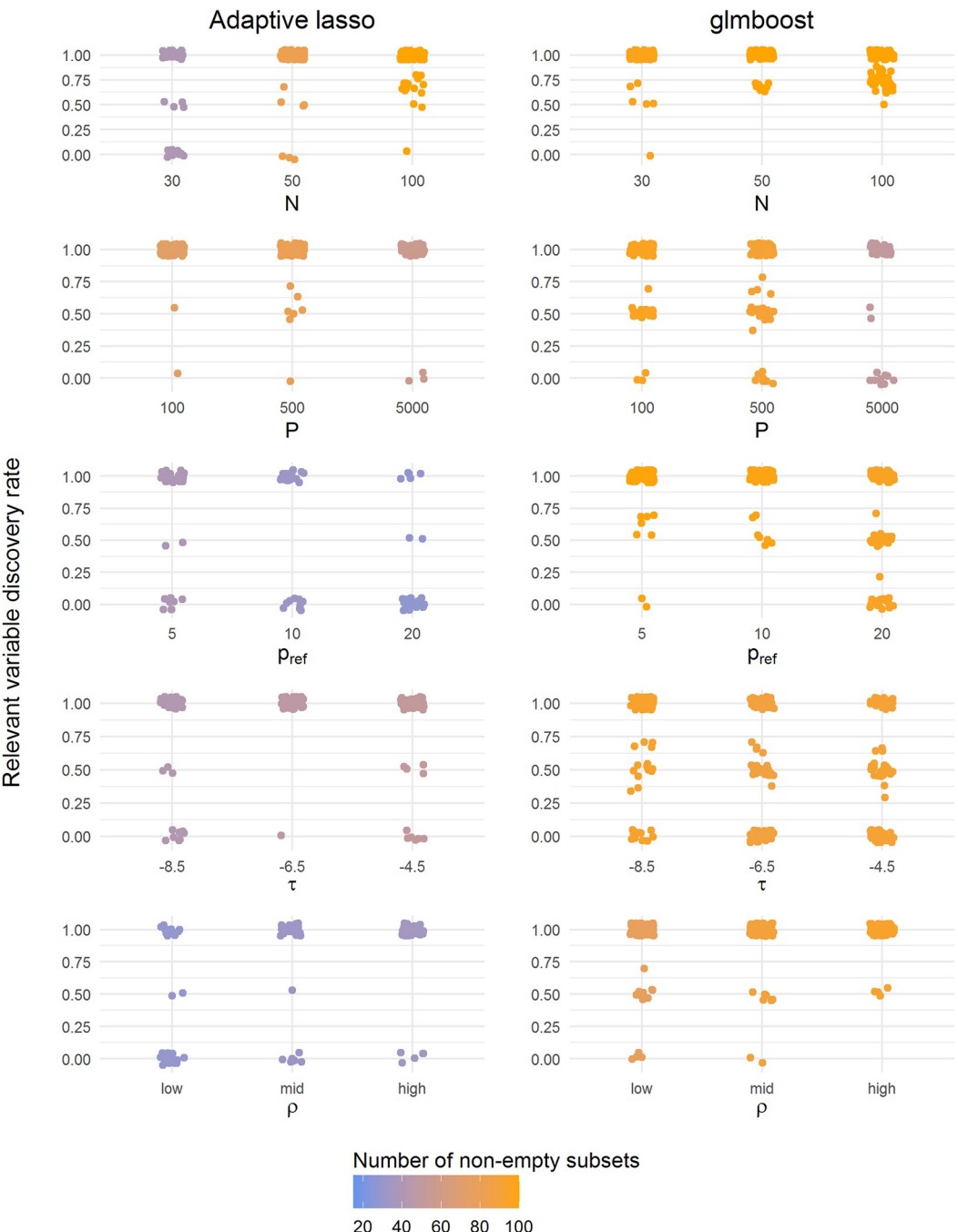

**Fig 4. Summary of RVDR of each subset in all scenarios using adaptive lasso and glmboost.** The colors represent the number of subsets resulting in at least one stable variable (blue: Low, orange: High). A total number of 100 realizations of simulated datasets were drawn for each scenario.

Fig 6 summarizes the numbers of selected base-learners among the 40 subsamples in each scenario. Both glmboost and gamboost drastically reduced the number of base-learners, from 333 and 663 to less than 20 predictors in each trained model (see S1 Table for details). Although gamboost had more choices for selecting base-learners than glmboost, the average number of selected base-learners in both models were similar in every scenario. The unstable

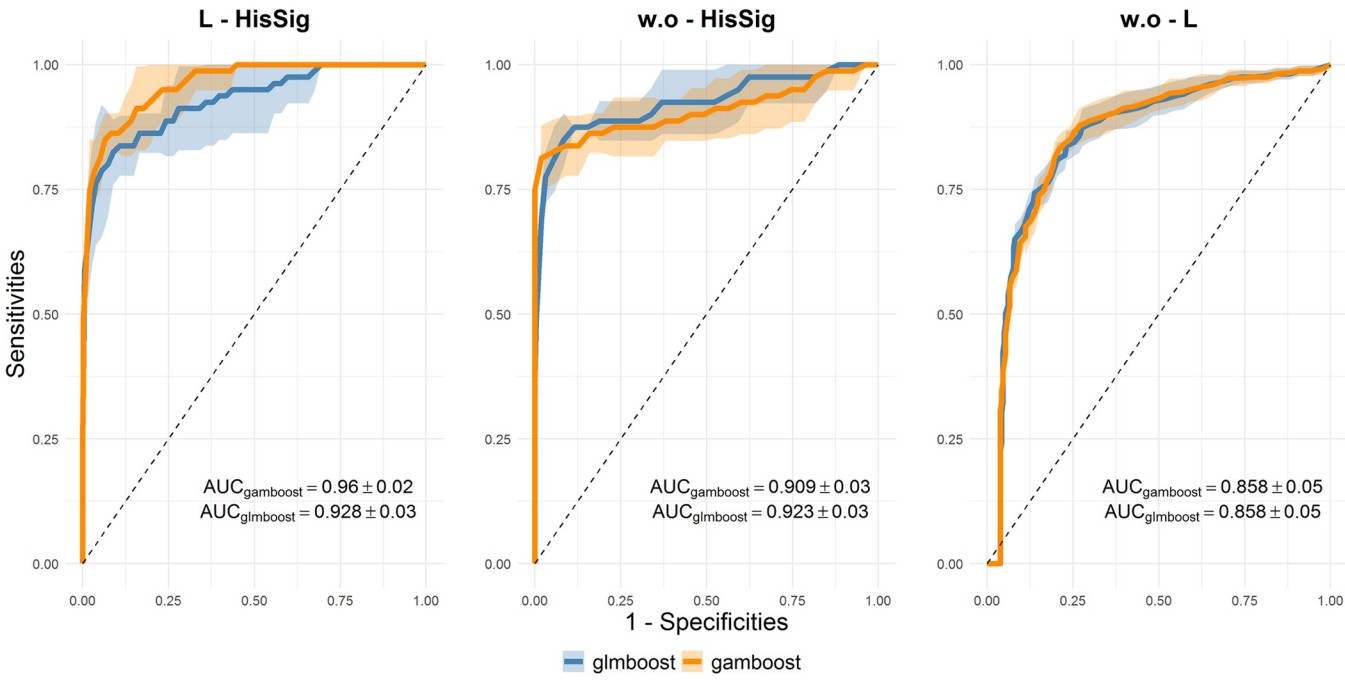

**Fig 5. Averaged ROC curves of the prediction performance glmboost (blue) and gamboost (orange).** The averaged ROC curves were estimated based on 40 subsamples generated by the repeated stratified 4-fold cross-validation in the three comparisons of the FaPaCa sample. The shaded areas represent the one standard deviation intervals. At the right bottom corner, the average AUCs and their standard deviations are shown.

variable selection performance of glmboost in scenario L-HisSig may explain the suboptimal test prediction performance.

Next, we performed CPSS to identify subsets of the stable biomarkers and to control the number of potential false discovered biomarkers. Fig 7 depicts the CPSS results of $B = 50$ subsampling procedures of glmboost and gamboost. Not surprisingly, only a few base-learners were observed above the stability threshold $\pi_{thr}$. Even though more stable variables were selected via gamboost shown in Fig 7, the resulting stable subsets of both models were similar: all stable biomarkers selected by glmboost were also included in the stable subsets of gamboost. In addition, in the scenario L-HisSig, while the selection probability of PLA2G7 by glmboost was marginally below the threshold, gamboost identified PLA2G7 as a stable biomarker, demonstrating the consistency of the results with glmboost and gamboost. In scenario w.o-L, both linear and centered smooth p-spline base-learners of PCSK9 were included in the stable subset, indicating that PCSK9 exhibited a statistically relevant non-linear relationship between patients without and with visible pancreatic lesions. It is worth noting that most of the selected stable biomarkers by gamboost are linear base-learners, and only two of them were embedded with the p-spline base-learner, although gamboost offered a more flexible model design than glmboost.

## Discussion

### Stability selection controls false discovery rate of variables

Discovering prognostic biomarkers and building prediction models are the primary goals of most 'omics studies. Although high-throughput proteomics screening allows us to explore comprehensive biomarker profiles easily, this innovative analytical technique comes along with a high-dimensional resulting data structure, which is challenging for statistical analysis.

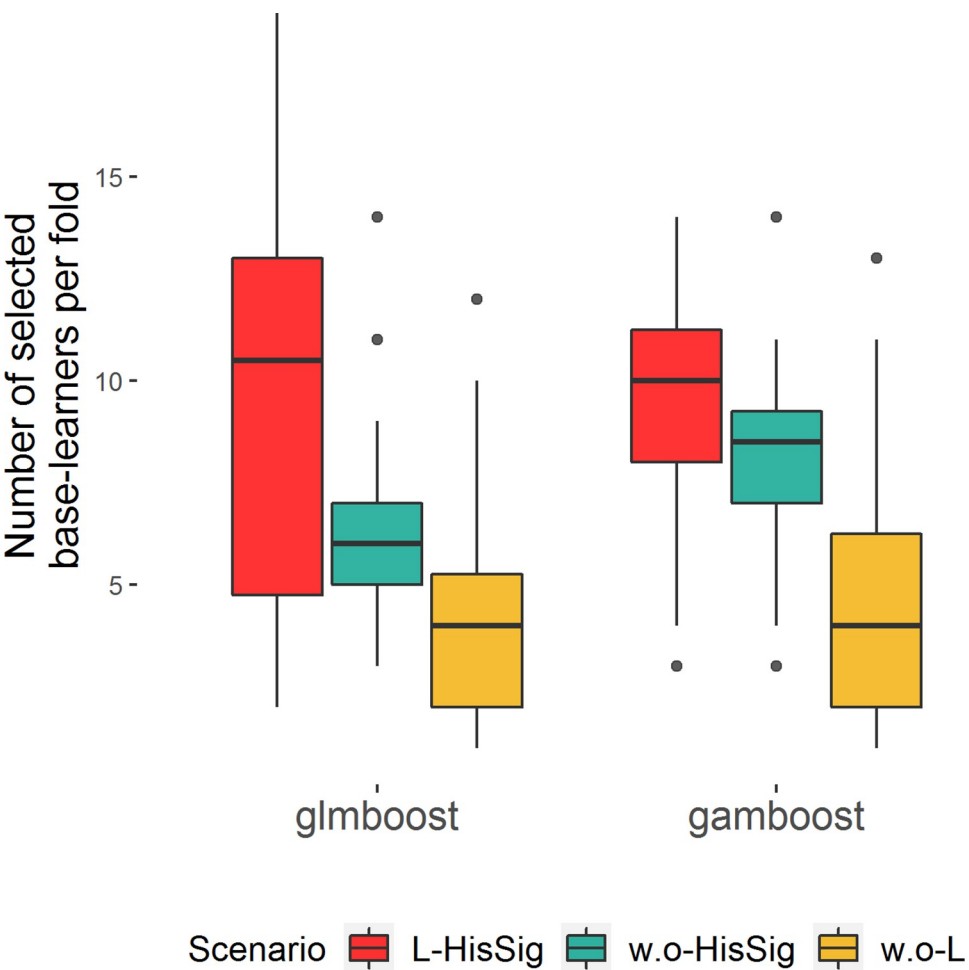

**Fig 6. Number of selected assays by glmboost and gamboost in all scenarios.** Each boxplot summarizes the results of 40 subsamples generated by repeated stratified 4-fold cross-validation in the respective scenario L-HisSig (red), w.o-HisSig (green) and w.o-L (yellow).

In our first simulation, despite the small sample sizes, dimensional reduction via variable selection could enhance model prediction performance in high-dimensional data: both glmboost and adaptive lasso achieved higher averaged AUCs than ridge regression (Fig 1). Moreover, variable selection improved the interpretability of fitted models by reducing the number of dimensions. For example, in the FaPaCa study, the number of selected stable biomarkers was mostly below 20 in all scenarios (Fig 6). The results of the simulation and FaPaCa studies demonstrated that the iterative model-fitting approach of mboost is a robust method in the context of analysing small, unbalanced, and high-dimensional datasets. Moreover, the flexible model architecture of gamboost enables us to include non-linear effects into the models and gain insight into the non-linear relationship of the biomarkers when comparing different phenotypes.

Nevertheless, the instability of selected variable subsets was observed in adaptive lasso, glmboost, and gamboost in both simulation and FaPaCa studies (Figs 2 and 6), meaning that the applied variable selection methods were sensitive to the partitions of the samples and the sample sizes. The unbalanced data structure also contributed to the instability of variable selection due to the unequal information between the binary classes. Therefore, in addition to the

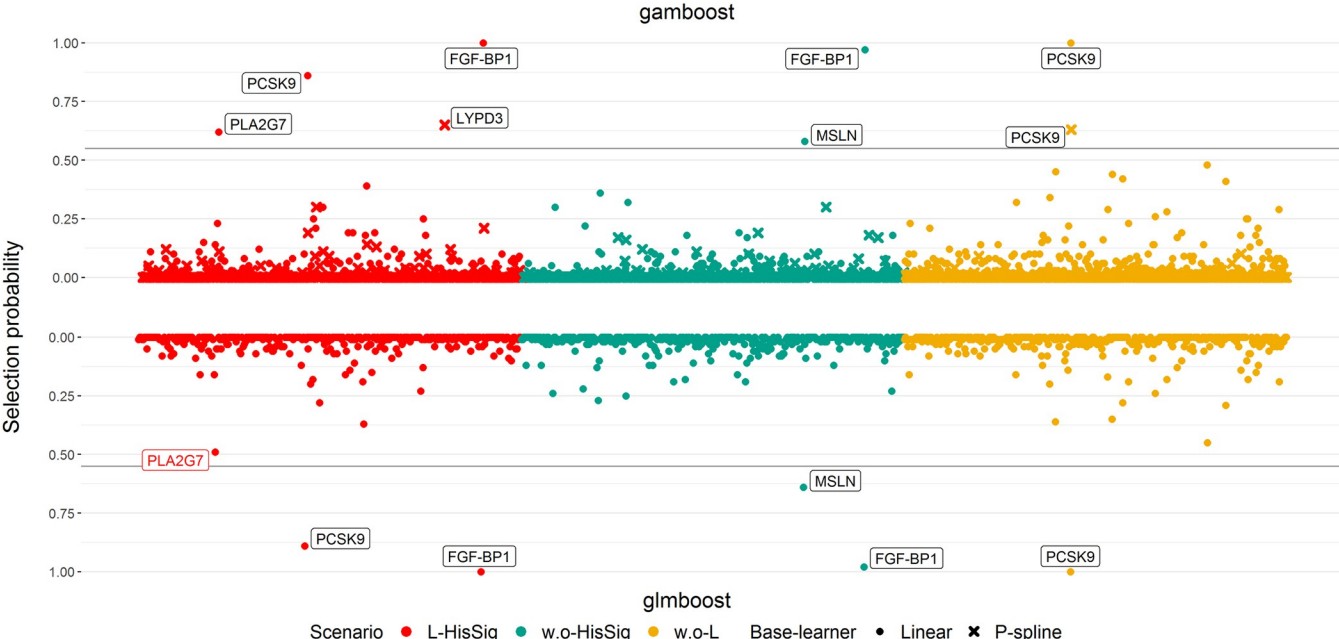

**Fig 7. Summary of the stability selection results using glmboost and gamboost.** The stability selection results using glmboost (bottom) and gamboost (top) among the scenarios L-HisSig (red), w.o-HisSig (green) and w.o-L (yellow). The grey line represents the corresponding cut-off level under the assumption of a unimodal distribution. The assays with selection probability higher than (black font) or slightly below the cut-off (red font) are annotated. The results of stability selection are summarized in S2 Table.

variable selection methods, applying stability selection to investigate the stability of the selected variables has been essential in our studies. According to the results of our second simulation, CPSS can effectively govern the number of falsely selected variables and hence identify a stable subset. Most of the variables in the stable subsets chosen by CPSS with adaptive lasso and glmboost were relevant variables (Fig 4), although not all the relevant variables were selected (Fig 3). The highly intercorrelated structure of the relevant variables could be one of the underlying reasons why not all relevant variables were selected. Compared to adaptive lasso, CPSS worked better with glmboost in that more variables and more relevant variables were included into the stable subsets. In the FaPaCa study, only a few biomarkers selected by glmboost and gamboost were qualified as stable biomarkers (Fig 7), and many of the selected biomarkers (summarized in Fig 6) were potentially false positive discoveries. Gamboost took advantage of the added non-linear effects, and CPSS selected two extra stable biomarkers, LYPD3 and PCSK9, modelled with smooth p-spline base-learners. We additionally performed CPSS with the adaptive lasso in the FaPaCa study, and, surprisingly, adaptive lasso reproduced very similar results compared to glmboost and gamboost regarding the selected variables (S1–S4 Figs).

To summarize, both applied statistical learning methods, glmboost and gamboost, could provide promising results in small and unbalanced datasets, and stability selection helped us screen out the false positive biomarkers. We recommend that stability selection should be included as a standard procedure of biomarker discovery in 'omics studies to prevent false positive findings.

## Post-selection inference

The notable advantage of using model-based variable selection methods is that they share the same interpretation as the classical statistical models. Therefore, we can easily infer the

**Table 1. The estimated odds ratios of FGF-BP1, PCSK9, PLA2G7, and MSLN via linear base-learners in each scenario.**

| Scenario<br>Biomarkers | L-HisSig | w.o-HisSig | w.o-L |
|---|---|---|---|
| FGF-BP1 | 4.22 (3.12, 15.93) | 43.11 (9.29, 162.98) | Not selected |
| PCSK9 | 0.61 (0.27, 0.94) | Not selected | 3.23 (1.35, 49.77) |
| PLA2G7 | 0.49 (0.06, 0.67) | Not selected | Not selected |
| MSLN | Not selected | 5.91 (1.34, 14.01) | Not selected |

individual effects of the biomarkers or their corresponding base-learners from the fitted models. We performed post-selection inference on the resulting subset of stable biomarkers from the gamboost-embedded CPSS (Fig 7) by, in each scenario, fitting only the corresponding stable biomarkers to the labelled statuses using gamboost again. Further details of the model fitting are described in the S1 and S2 Appendices. Table 1 shows the estimated odds ratios of the linear base-learners of the stable variables in the respective scenarios (the non-linear p-spline base-learners have been omitted at this step for a better interpretability). It should be noted that the odds ratio refers to an increase of 1 NPX of each assay. The high estimated odds ratios of FGF-BP1 in scenarios w.o-HisSig and L-HisSig may indicate that FGF-BP1 could be a

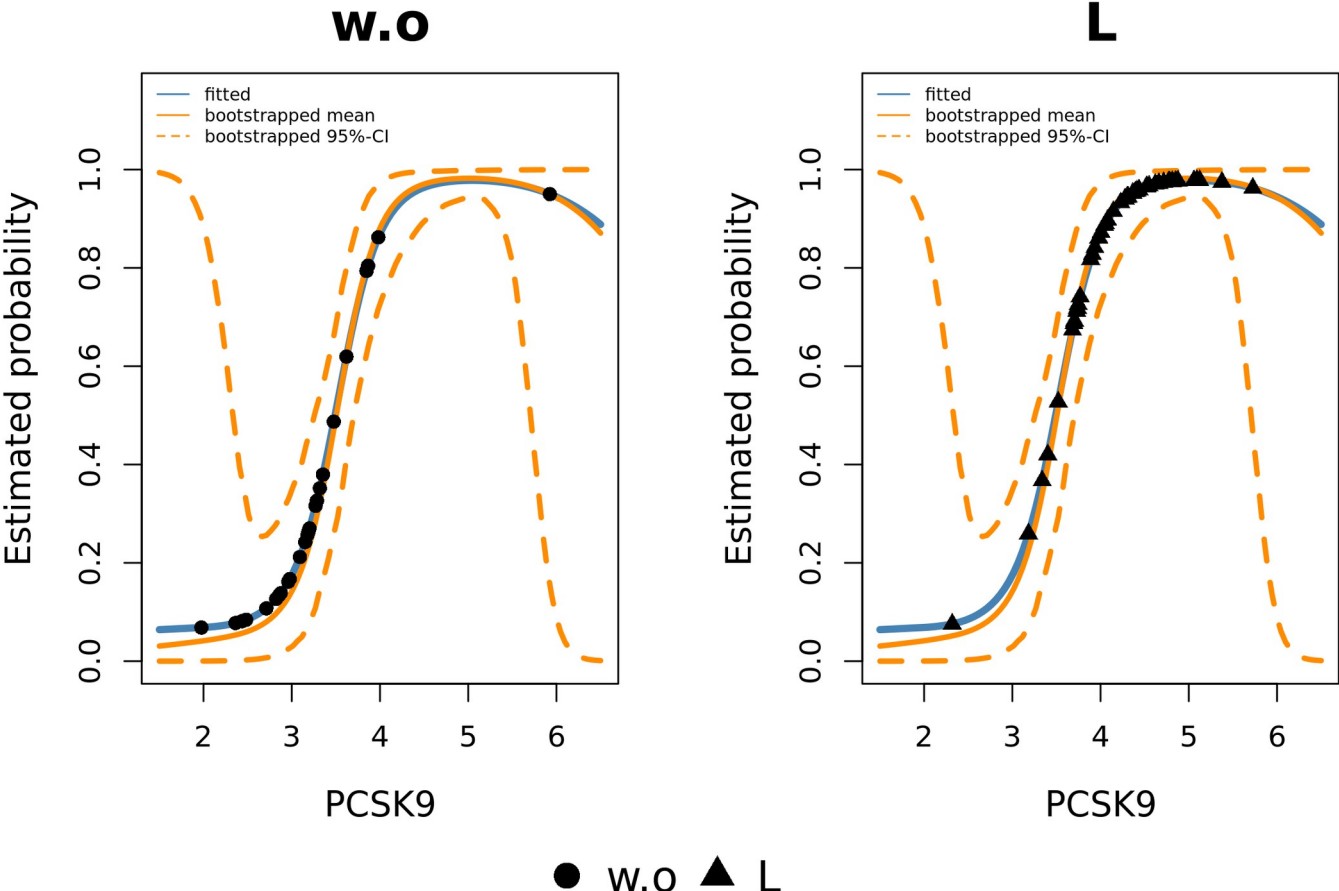

**Fig 8. The estimated probability of being classified as 'L' in scenario w.o-L using PCSK9 with gamboost.** Blue and orange solid lines represent the prediction results of the fitted gamboost and the bootstrapped mean prediction results, respectively. The dotted lines describe the bootstrapped 95% confidence interval. The left plot shows the observation of status w.o and the right plot the observation of status L.

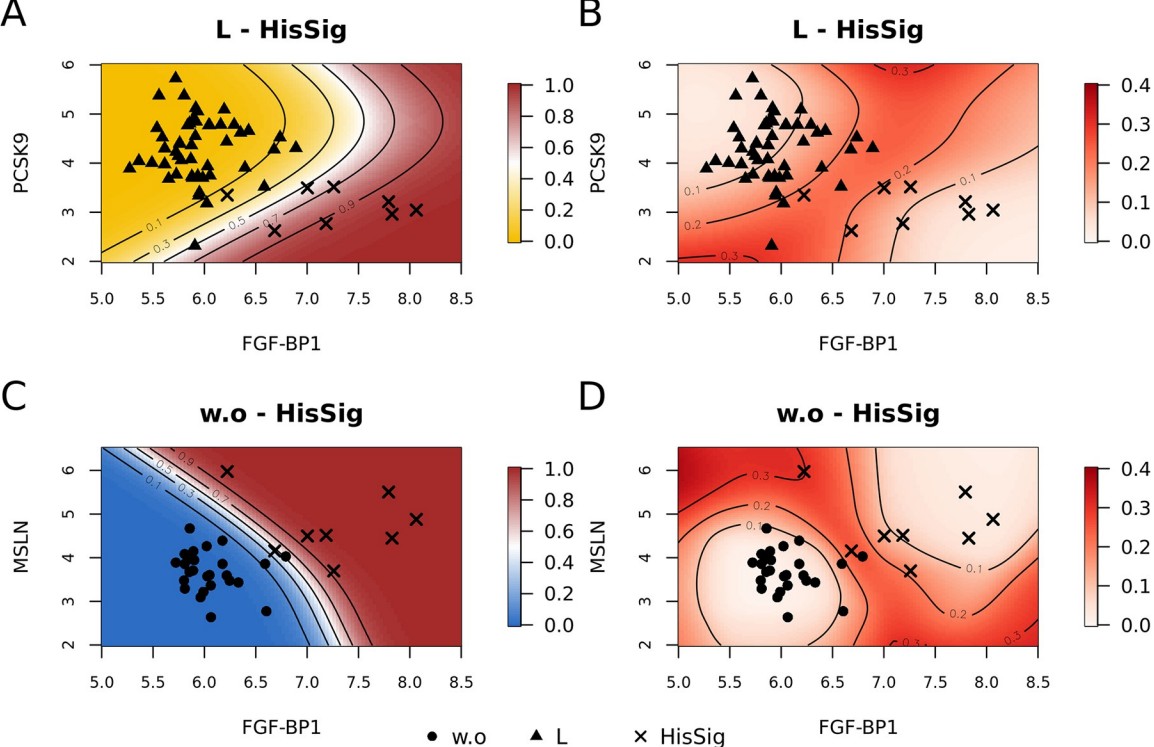

**Fig 9. The estimated classification results and the standard errors using gamboost in scenarios L-HisSig and w.o-HisSig.** (A) and (C): The estimated classification results by gamboost in scenarios L-HisSig and w.o-HisSig. The colors represent the tendency of prediction results for the lesion status (blue = w.o, yellow = L and brown = HisSig). The contours indicate the estimated response with an interval width of 0.2. (B) and (D): The estimated standard errors of classification results in both scenarios obtained by bootstrapping. Red regions represent high estimated standard errors (low certainty). The contours represent the estimated standard errors with an interval width of 0.1.

potential biomarker for detecting the development of histologically significant lesions in IARs. Fig 8 depicts the prediction results of gamboost in scenario w.o-L and shows that an increase in PCSK9 concentration was associated with higher odds having lesions versus not having lesions. These results indicate that an early stage of the physiological development of pre-cancerous tissue may be characterized by increased PCSK9 levels. In contrast, the progression from lesions to histologically significant lesions was associated with a lower level of PCSK9 according to the estimated odds ratio < 1 in Table 1.

The values in brackets are 90% bootstrapped confidence intervals of the estimated odds ratios.

In addition to univariate interpretations, we visualized the marginal classification results of the stable variables. Fig 9A and 9C depict the conditional prediction results of scenarios L-His-Sig and w.o-HisSig. Fig 9B and 9D show the uncertainty of the prediction results in terms of standard errors estimated via 1000 bootstrapped resamples. According to the prediction surfaces of Fig 9A and 9C, the corresponding stable biomarkers in both scenarios were able to produce promising classification results, demonstrated by the clear boundary between statuses. Furthermore, the high certainty of the estimated classification results, represented by the white regions in Fig 9B and 9D, shows the robustness of estimated effects against the resampling. The regions with less observations (red regions) are generally characterized by low certainty. Most observations are located at the high certainty regions, but a few are found in the low

certainty regions, especially those around the 50% prediction boundary and those far away from the majority of observations. This shows that extra information and data are necessary to achieve a precise classification result for the patients in the red regions in both scenarios. S5 and S6 Figs depict the mean, 2.5%-quantile and 97.5%-quantile of the bootstrapped prediction results of scenarios L-HisSig and w.o-HisSig.

## Biological relevance of found stable biomarkers

Some previous studies showed that the discovered stable biomarkers in the FaPaCa study play a relevant role in various cancer developments. First, FGF-BP1 plays a crucial role in regulatory factors through binding directly to FGF-1 and FGF-2. Therefore, it boosts their biological activities, such as signaling, cell proliferation and angiogenesis [43,44]. Furthermore, overexpression of FGF-BP1 was found in pancreatic and colon cancer [45]. Second, MSLN was identified as a tumor-differentiation antigen with an overexpression discovered in multiple cancer types, such as epithelial mesothelioma, pancreatic cancer, and lung adenocarcinoma. It was shown to be a prognostic biomarker of early cancer-specific mortality of PDAC [46]. Third, Vainio et al. found that PLA2G7 promotes the cell migration of prostate cancer [47]. Fourth, some studies discussed the association of LYPD3 with the progression of PDAC, melanoma and non-small cell lung cancer [48–51]. Finally, PCSK9 may indirectly influence cancer growth via its cholesterol-regulating function, as the low-density-lipoprotein cholesterol supports tumor development [52,53]. Li et al. also proposed that immune checkpoint therapy for cancer may benefit from inhibition of PCSK9 [54,55].

In addition to the biological significance of individual biomarkers, we performed an overrepresentation analysis to explore the biological pathways of the discovered stable protein biomarkers. S3 Table shows nine statistically significant biological pathways, i.e. adjusted p-value < 0.05, found in the Reactome pathway database [37]. Three of the nine pathways involve the stable biomarker pairs LYPD3-MSLN, providing convincing evidence of their close biological relationships and potential biological interaction.

## Strengths and limitations

Compared to other studies on familial pancreatic cancer, our study investigated a very rare and valuable dataset of high-throughput proteomics. Histologically significant lesions can be interpreted as the last precursor stage before invasive PDAC [12,19–21]. A well-guided decision to resect the pancreas at the latest possible timepoint, but still prior to the development of PDAC, has the potential to have a major impact on survival and quality of life of FPC patients. Unfortunately, little data is available on IARs prior to the onset of PDAC. We consider it a major strength of our study that we had information about different pre-cancer phenotypes in IARs, which allowed us to investigate potential pathophysiological mechanisms of cancer onset. However, the special design and small sample sizes introduces several limitations with regard to statistical modelling. Hence, we reduced the flexibility of statistical learning methods to maintain the interpretability of the results and introduced an additional layer of stability selection to identify plausible and stable subsets of biomarkers. Furthermore, we conducted simulation studies for a thorough investigation of the validity and robustness of our findings. We consider our analyses to be an important hypothesis-generating step, to be succeeded by several subsequent clinical evaluation and validation procedures. For future research, it would be of interest to define a semi-supervised screening procedure that integrates prior clinical knowledge about protein networks, allows prioritizing variables within known clusters, and defines a subset of biomarkers that is hard-wired to be included in the model.

## Conclusions

Our work focused on applying advanced statistical learning algorithms to discover potentially relevant protein biomarkers relating to the development of PDAC. The proposed statistical models joined the protein selecting and model-fitting procedures such that the proteins were chosen based on the model prediction performance. Results from the simulation study and FaPaCa study both show that the adaptive lasso and iterative fitting process using mboost could deal with the small, unbalanced, and high-dimensional data and identify a compact subset of protein biomarkers. We also demonstrated how stability selection identified the subsets of stable protein biomarkers that are robust in the context of the unbalanced study design. With PCSK9, FGF-BP1, PLA2G7, LYPD3, and MSLN, we identified five potentially important proteins in the process of development of PDAC. Our data suggest that the latter four proteins play an important role in the progression of lesions to histologically significant lesions, while PCSK9 appears to be more important in forming early lesions. This discovery provides us with a clear direction for further investigations and more precise clinical hypotheses regarding the development of FPC and optimal strategies for its early detection.

## Supporting information

**S1 Fig. Averaged ROC curves of the prediction performance adaptive lasso, glmboost, and gamboost.** Averaged ROC curves of the prediction performance adaptive lasso (yellow), glmboost (blue), and gamboost (orange) are estimated based on 40 subsamples generated by the repeated stratified 4-fold cross-validation in the three scenarios of the FaPaCa study. The shaded areas represent the one standard deviation intervals. At the right bottom corner, the averaged AUCs and their standard deviations are shown.
(DOCX)

**S2 Fig. Number of selected variables/base-learners by adaptive lasso, glmboost, and gamboost.** Each boxplot summarizes the results of 40 subsamples generated by repeated stratified 4-fold cross-validation in each fold in the respective scenario L-HisSig (red), w.o-HisSig (green) and w.o-L (yellow).
(DOCX)

**S3 Fig. Summary of the stability selection results using adaptive lasso and gamboost.** The stability selection results using adaptive lasso (bottom) and gamboost (top) among the scenarios L-HisSig (red), w.o-HisSig (green), and w.o-L (yellow). The grey line represents the corresponding cut-off level under the assumption of a unimodal distribution. The assays with selection probability higher than (black font) or slightly below the cut-off (red font) are annotated.
(DOCX)

**S4 Fig. Summary of the stability selection results using adaptive lasso and glmboost.** The stability selection results using adaptive lasso (bottom) and glmboost (top) among the scenarios L-HisSig (red), w.o-HisSig (green), and w.o-L (yellow). The grey line represents the corresponding cut-off level under the assumption of a unimodal distribution. The assays with selection probability higher than (black font) or slightly below the cut-off (red font) are annotated.
(DOCX)

**S5 Fig. The estimated classification results of gamboost in scenario L-HisSig.** The classification results are conditioned on a low concentration of PLA2G7 and a high concentration of LYPD3. (A) depicts the prediction results of the fitted gamboost model. (B), (C), and (D)

represent the mean, lower quantile (2.5%), and upper quantile (97.5%) of the prediction results estimated via bootstrapping.
(DOCX)

**S6 Fig. The estimated classification results of gamboost in scenario w.o-HisSig.** (A) depicts the prediction results of the fitted gamboost model. (B), (C), and (D) represent the mean, lower quantile (2.5%), and upper quantile (97.5%) of the prediction results estimated via bootstrapping.
(DOCX)

**S1 Table. The results of variable selection using glmboost and gamboost in the FaPaCa study.** This table summarizes the results of the 10 times repeated 4-fold cross validation in terms of the selection frequency of all Olink biomarkers in each scenario and each mboost model (glmboost and gamboost).
(XLSX)

**S2 Table. The results of stability selection using glmboost and gamboost in the FaPaCa study.** This table summarizes the results of the stability selection in terms of the selection frequency of all Olink biomarkers in each scenario and each mboost model (glmboost and gamboost). The asterisked biomarkers are regarded as stable biomarkers.
(XLSX)

**S3 Table. The results of over-representation analysis using Reactome pathway database.**
(DOCX)

**S1 Appendix. Supplementary methods.**
(DOCX)

**S2 Appendix. Rmarkdown of data analysis in simulation study and FaPaCa study.**
(DOCX)

## Acknowledgments

The authors wish to thank the families of the FaPaCa registry for their participation, N. Gercke for excellent technical assistance, and A. Hall for critical reading of the manuscript.

## Author Contributions

**Conceptualization:** Chung Shing Rex Ha, Martina Müller-Nurasyid, Detlef K. Bartsch, Emily P. Slater, Konstantin Strauch.

**Data curation:** Chung Shing Rex Ha, Martina Müller-Nurasyid, Emily P. Slater.

**Formal analysis:** Chung Shing Rex Ha, Martina Müller-Nurasyid, Federico Marini, Konstantin Strauch.

**Funding acquisition:** Detlef K. Bartsch, Emily P. Slater, Konstantin Strauch.

**Investigation:** Chung Shing Rex Ha, Martina Müller-Nurasyid, Agnese Petrera, Stefanie M. Hauck, Emily P. Slater, Konstantin Strauch.

**Methodology:** Chung Shing Rex Ha, Martina Müller-Nurasyid, Agnese Petrera, Stefanie M. Hauck, Federico Marini, Konstantin Strauch.

**Project administration:** Martina Müller-Nurasyid, Detlef K. Bartsch, Emily P. Slater, Konstantin Strauch.

**Resources:** Detlef K. Bartsch, Emily P. Slater.

**Supervision:** Martina Müller-Nurasyid, Konstantin Strauch.

**Validation:** Chung Shing Rex Ha, Martina Müller-Nurasyid, Federico Marini, Konstantin Strauch.

**Visualization:** Chung Shing Rex Ha.

**Writing – original draft:** Chung Shing Rex Ha, Martina Müller-Nurasyid, Emily P. Slater, Konstantin Strauch.

**Writing – review & editing:** Chung Shing Rex Ha, Martina Müller-Nurasyid, Agnese Petrera, Stefanie M. Hauck, Federico Marini, Detlef K. Bartsch, Emily P. Slater, Konstantin Strauch.

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
