## [Decision Letter · Decision Letter 0]

22 Jun 2022

PONE-D-22-07771Proteomics biomarker discovery for individualized prevention of familial pancreatic cancer using statistical learningPLOS ONE

Dear Dr. Rex Ha,

Thank you for submitting your manuscript to PLOS ONE. After careful consideration, we feel that it has merit but does not fully meet PLOS ONE’s publication criteria as it currently stands. Therefore, we invite you to submit a revised version of the manuscript that addresses the points raised by the reviewer during the review process.

We look forward to receiving your revised manuscript.

Kind regards,

Surinder K. Batra

Academic Editor

PLOS ONE

Journal Requirements:

“This research was supported by a grant from the Wilhelm Sander-Stiftung to EPS, DKB, and KS (No. 2018.022.1) and a generous donation from the GAUFF-Foundation. Furthermore, this research was supported within the Munich Center of Health Sciences (MC-Health), Ludwig Maximilian University of Munich, as part of LMUinnovativ.”  

5. Please include a copy of Table 1 which you refer to in your text on page 16.

Reviewers' comments:

Reviewer's Responses to Questions

**Comments to the Author**

1. Is the manuscript technically sound, and do the data support the conclusions?

Reviewer #1: Yes

2. Has the statistical analysis been performed appropriately and rigorously? 

Reviewer #1: I Don't Know

3. Have the authors made all data underlying the findings in their manuscript fully available?

Reviewer #1: No

4. Is the manuscript presented in an intelligible fashion and written in standard English?

Reviewer #1: Yes

5. Review Comments to the Author

Reviewer #1: This reviewer has expertise only in biological aspects of the manuscript. Biological part seems to be very confusing, couple biomarkers are mentioned, others are not. There is no table summarizing all stable biomarkers discovered by O-Link. 20-biomarker panels are mentioned, but these biomarkers are not shown. In the introduction TIMP1 and LCN2 are mentioned. Were these biomarkers discovered by O-Link? What were other stable biomarkers discovered by O-Link, and what biological pathways do they represent? The biological part needs to be re-written for clarity and to better delineate biological significance of stable biomarkers.

6. PLOS authors have the option to publish the peer review history of their article (what does this mean?). If published, this will include your full peer review and any attached files.

Reviewer #1: No

---

## [Author Response · Author response to Decision Letter 0]

19 Aug 2022

Response to Reviewers

Journal Requirements:

To Academic Editor: We would like to thank Prof. Batra for his effort in managing our submission and helpful comments. We have addressed the questions raised by Prof. Batra as follows:

Authors’ reply:

Thank you for the kind reminder. We have aligned our manuscript according to the PLOS ONE’s style requirements.

Authors’ reply:

We thank the academic editor for highlighting the insufficient information in the submitted Data Availability Statement. As mentioned in our cover letter, we would like to include the Data Availability Statement in our manuscript as follows:

The individual-level data of the FaPaCa study are not publicly available, because the data contain sensitive patient data which underlie data protection rules. This is in accordance with the local ethic vote and the regulations of the FaPaCa registry. Patients’ characteristics are available upon request from the FaPaCa study registry (contact information: fapaca@med.uni-marburg.de). The aggregated results from the FaPaCa study and the full codes for the simulation are available in its Supporting Information files.

“This research was supported by a grant from the Wilhelm Sander-Stiftung to EPS, DKB, and KS (No. 2018.022.1) and a generous donation from the GAUFF-Foundation. Furthermore, this research was supported within the Munich Center of Health Sciences (MC-Health), Ludwig Maximilian University of Munich, as part of LMUinnovativ.” 

Authors’ reply:

Thank you for reviewing the financial disclosure and requesting the role of the funders in the study. We have written the Role of Funder statement as follows:

This statement has been included in the cover letter.

Authors’ reply:

Thank you for the reminder of including our full ethics statement. We have included the full ethics statement in the “Materials and Methods” section (line 142– 144), written as follows:

The FaPaCa registry, including the genetic analyses and the screening program, was approved by the Ethics Committee of the Philipps University of Marburg (No. 36/1997, last amendment 9/2010), and all participants provided written informed consent.

5. Please include a copy of Table 1 which you refer to in your text on page 16.

Authors’ reply:

Thank you for the reminder of including a copy of Table 1. We have attached the Table 1 in the manuscript.

Reviewers' comments:

To Reviewer: We would like to thank the reviewer for the effort of reading and reviewing our manuscript. Also, we appreciate the reviewer’s comments and suggestions, providing an insightful perspective on the biological interpretation of our findings and helping us improve the quality of the manuscript. Therefore, we have responded to the reviewer’s comments and critique as follows:

1. Is the manuscript technically sound, and do the data support the conclusions?

Reviewer #1: Yes

2. Has the statistical analysis been performed appropriately and rigorously?

Reviewer #1: I Don't Know

Authors’ reply:

Thank you for sharing your opinions on our statistical analysis. The analysis approach we used in the manuscript is a statistical learning method - model-based boosting. It helped us to identify useful prognostic biomarkers in comprehensive protein profiles obtained by high-throughput proteomics. Additionally, we implemented another statistical method - stability selection - to identify a set of statistically stable biomarkers among the above-mentioned useful prognostic biomarkers. The set of statistically stable biomarkers refers to the set of biomarkers with a robust prognostic power. The technical details of the statistical modelling are demonstrated in the Rmarkdown file attached in the Supporting Information (S2 Appendix).

3. Have the authors made all data underlying the findings in their manuscript fully available?

Reviewer #1: No

Authors’ reply:

Thank you for pointing out that we did not provide the data underlying the finds in our manuscript. Owing to medical privacy and sensitive patient information, the data of the FaCaPa study are not publicly available, but patients’ characteristics are available upon request. Instead, the data in simulation study are available in the supporting information files. The data availability statement in the manuscript has been updated as follows:

The individual-level data of the FaPaCa study are not publicly available, because the data contain sensitive patient data which underlie data protection rules. This is in accordance with the local ethic vote and the regulations of the FaPaCa registry. Patients’ characteristics are available upon request from the FaPaCa study registry (contact information: fapaca@med.uni-marburg.de). The aggregated results from the FaPaCa study and the full codes for the simulation are available in its Supporting Information files. 

4. Is the manuscript presented in an intelligible fashion and written in standard English?

Reviewer #1: Yes 

5. Review Comments to the Author

Reviewer #1: This reviewer has expertise only in biological aspects of the manuscript. Biological part seems to be very confusing, couple biomarkers are mentioned, others are not. There is no table summarizing all stable biomarkers discovered by O-Link. 

Authors’ reply: 

Thank you for commenting on our manuscript in biological aspects and mentioning our unclear explanations in the biological part. 

First, we would like to emphasize and clarify the concept of stable biomarkers. In our manuscript, stable biomarkers refer to the statistically stable biomarkers discovered by our statistical models, instead of biologically stable biomarkers. 

We have created a summary table, including all biomarkers obtained via Olink and the corresponding resulting selection frequency in the stability selection analysis. All stable biomarkers have been asterisked in the table. The table has been attached in this document and in the Supporting Information, labelled as S1 Table.

20-biomarker panels are mentioned, but these biomarkers are not shown. 

Authors’ reply: 

The summary of selected biomarkers referring to Figure 6 is now also attached in this document and in the Supplementary Information, labelled as S2 Table.

In the introduction TIMP1 and LCN2 are mentioned. Were these biomarkers discovered by O-Link? 

Authors’ reply: 

As described in the corresponding publications Slater et al. 2013 and Bartsch et al. 2018, TIMP1 and LCN2 were discovered via the enzyme-linked immunosorbent assay (ELISA), instead of Olink.

What were other stable biomarkers discovered by O-Link, and what biological pathways do they represent? The biological part needs to be re-written for clarity and to better delineate biological significance of stable biomarkers.

Authors’ reply:

Thank you for mentioning the insufficient biological part about biological pathways and significance of discovered stable biomarkers. We have performed an overrepresentation analysis using the Reactome pathway database on the stable biomarkers (FGF-BP1, MSLN, PCSK9, PLA2G7 and LYPD3). The results are summarized in S3 Table in the manuscript, also attached in this document. We have also rephrased the biological part (line 455 - 473 on p.17 & 18) in the manuscript, and addressed the stable biomarkers’ biological significance, shown as follows:

Biological relevance of found stable biomarkers

Some previous studies showed that the discovered stable biomarkers in the FaPaCa study play a relevant role on various cancer developments. First, FGF-BP1 plays a crucial role on regulatory factors through binding directly to FGF-1 and FGF-2. Therefore, it boosts their biological activities, such as signaling, cell proliferation and angiogenesis [35], [36]. Furthermore, an overexpression of FGF-BP1 was found in pancreatic and colon cancer [37]. Second, MSLN was identified as a tumor-differentiation antigen with an overexpression discovered in multiple cancer types, such as epithelial mesothelioma, pancreatic cancer, and lung adenocarcinoma. It was shown to be a prognostic biomarker of early cancer-specific mortality of PDAC [38]. Third, Vainio et al. found that PLA2G7 promotes the cell migration of prostate cancer [39]. Fourth, some studies discussed the association of LYPD3 with progression of PDAC, melanoma and non-small cell lung cancer [40]–[43]. Finally, PCSK9 may have an indirect influence on cancer growth via its cholesterol-regulating function, as the low-density-lipoprotein cholesterol supports tumor development [44], [45]. Li et al. also proposed that immune checkpoint therapy for cancer may benefit from inhibition of PCSK9 [46], [47]. 

In addition to the biological significance of individual biomarkers, we performed an over-representation analysis to explore the biological pathways of the discovered stable protein biomarkers. S3 Table shows 9 statistically significant biological pathways, i.e. adjusted p-value < 0.05, found in the Reactome pathway database [29]. Three of the pathways involve the stable biomarker pairs LYPD3-MSLN, providing convincing evidence on their close biological relationships and potential biological interaction.

We also added the following paragraph to the Material and Methods (line 236 – 240 on p.9) section:

To identify the biological pathway of the resulted subset of stable biomarkers, we performed an over-representation analysis using the Reactome pathway database [29]. In the analysis, all 330 eligible Oink biomarkers were used as background genes, but only 286 were recognized by Reactome. Fisher’s exact test was used to determine the statistical significance of the pathways and the Benjamini-Hochberg procedure was applied to control the false discovery rate. The significance level was set at 5%.

We adjusted the section on Statistical software (line 244 – 247 on p.10) as follows:

All statistical analyses were performed in the statistical programming language R (version 3.6.3) [30]. We performed ridge regression and adaptive lasso with package glmnet [31], mboost with package mboost [25], stability selection with package stabs [27], and over-representation analysis with package clusterProfiler and ReactomePA [32]–[34].

6. PLOS authors have the option to publish the peer review history of their article (what does this mean?). If published, this will include your full peer review and any attached files.

Do you want your identity to be public for this peer review? For information about this choice, including consent withdrawal, please see our Privacy Policy.

Reviewer #1: No

---

## [Decision Letter · Decision Letter 1]

14 Nov 2022

PONE-D-22-07771R1Proteomics biomarker discovery for individualized prevention of familial pancreatic cancer using statistical learningPLOS ONE

Dear Dr. Ha,

Thank you for submitting your manuscript to PLOS ONE. After careful consideration, we feel that it has merit but does not fully meet PLOS ONE’s publication criteria as it currently stands. Therefore, we invite you to submit a revised version of the manuscript that addresses the points raised below by the reviewers  during the review process.

We look forward to receiving your revised manuscript.

Kind regards,

Surinder K. Batra

Academic Editor

PLOS ONE

Reviewers' comments:

Reviewer's Responses to Questions

**Comments to the Author**

1. If the authors have adequately addressed your comments raised in a previous round of review and you feel that this manuscript is now acceptable for publication, you may indicate that here to bypass the “Comments to the Author” section, enter your conflict of interest statement in the “Confidential to Editor” section, and submit your "Accept" recommendation.

Reviewer #1: (No Response)

Reviewer #2: All comments have been addressed

2. Is the manuscript technically sound, and do the data support the conclusions?

Reviewer #1: No

Reviewer #2: Yes

3. Has the statistical analysis been performed appropriately and rigorously? 

Reviewer #1: I Don't Know

Reviewer #2: Yes

4. Have the authors made all data underlying the findings in their manuscript fully available?

Reviewer #1: Yes

Reviewer #2: No

5. Is the manuscript presented in an intelligible fashion and written in standard English?

Reviewer #1: Yes

Reviewer #2: Yes

6. Review Comments to the Author

Reviewer #1: This is a potentially significant study and access to unique FaPaCa samples is a great plus. However, the main premise of the study, i.e., 'Proteomics biomarker discovery for individualized prevention of familial pancreatic cancer' is not supported by the choice of samples. It looks like all samples for this study were coming from PDAC-free individuals. The presence of a pre-neoplastic lesion does not necessarily translate into cancer. The manuscript needs advice from a pancreatic oncologist. Identified biomarkers need to be validated in another set.

Reviewer #2: The submitted manuscript by Ha et al applied advance simulation methods, like adaptive lasso to analyze the proteomic data to identify the promising biomarkers for the pancreatic cancer. The authors concluded that application of these advance learning methods improve the predictive accuracy of the proteomic data and provided the examples of FGF-BP1, PCSK9, and MSLN in pancreatic cancer inception and progression. Previous studies have also implicated FGF-BP1, and MSLN in pancreatic cancer pathology. Overall, the manuscript is well written and presented to understand the application of advance statistical methods. Moreover, authors addressed all the previous concerns.

7. PLOS authors have the option to publish the peer review history of their article (what does this mean?). If published, this will include your full peer review and any attached files.

Reviewer #1: No

Reviewer #2: No

---

## [Author Response · Author response to Decision Letter 1]

23 Dec 2022

Reviewers' comments:

To Reviewers: 

We would like to thank the reviewers for the effort of reading and reviewing our manuscript. Also, we appreciate the reviewers’ comments and suggestions, providing an insightful perspective on cancer pathology and helping us improve the quality of the manuscript. Therefore, we have responded to the reviewer’s comments and critique as follows:

Comments to the Author

1. If the authors have adequately addressed your comments raised in a previous round of review and you feel that this manuscript is now acceptable for publication, you may indicate that here to bypass the “Comments to the Author” section, enter your conflict of interest statement in the “Confidential to Editor” section, and submit your "Accept" recommendation.

Reviewer #1: (No Response)

Reviewer #2: All comments have been addressed

2. Is the manuscript technically sound, and do the data support the conclusions?

Reviewer #1: No

Reviewer #2: Yes

3. Has the statistical analysis been performed appropriately and rigorously?

Reviewer #1: I Don't Know

Reviewer #2: Yes

4. Have the authors made all data underlying the findings in their manuscript fully available?

Reviewer #1: Yes

Reviewer #2: No

5. Is the manuscript presented in an intelligible fashion and written in standard English?

Reviewer #1: Yes

Reviewer #2: Yes

6. Review Comments to the Author

Reviewer #1: This is a potentially significant study and access to unique FaPaCa samples is a great plus. However, the main premise of the study, i.e., 'Proteomics biomarker discovery for individualized prevention of familial pancreatic cancer' is not supported by the choice of samples. It looks like all samples for this study were coming from PDAC-free individuals. The presence of a pre-neoplastic lesion does not necessarily translate into cancer. The manuscript needs advice from a pancreatic oncologist. 

Our reply: 

Thank you very much for your review and advice. Our study aims to improve the detection of familial pancreatic cancer at its early, ideally pre-cancerous and still curable stage, with the help of protein biomarkers. Therefore, we focused on the individuals with precursor lesions, namely those with histologically significant lesions (HisSig), rather than individuals who already developed PDAC. High-grade pancreatic intraepithelial neoplasms (PanIN) and intraductal papillary-mucinous neoplasms (IPMN), which are well established obligate precancerous lesions (Ott et al. 2007, Permuth-Wey et al. 2009, Pittman et al. 2016, Makohon-Moore et al. 2018, Goggins et al. 2020), were found in the pancreas of the individuals in the HisSig group. Therefore, identifying and removing these lesions potentially can prevent the progression to PDAC. Since the visible pancreatic lesions on imaging cannot be reliably classified as benign or cancerous neither by endoscopic ultrasound (EUS) nor MRI, it is reasonable to assume that the IAR from FPC families and imagable pancreatic lesions (L group) are at risk for the development of PDAC in the future. We aim to provide an additional biomarker-based tool for regular screening of high-risk individuals to guide the decision for surgery with all the unfavourable consequences on life quality but with the major advantage of preventing the onset of PDAC. On the other hand, we agree that not all lesions progress to PDAC. Therefore, it is essential to identify novel biomarkers that can highlight persons with lesions on the path to progression to PDAC.

As a further aspect, we have opted to focus our analyses on individuals with precursor lesions rather than individuals with PDAC, since cancer development in the latter it is expected to have progressed to a more aggravating stage, at which already a larger portion of the proteomic and metabolic profile has been altered. Therefore, the results when contrasting patients with PDAC to unaffected individuals would most likely include biomarkers downstream of altered metabolic processes, e.g., caused by cachexia. (Dunne et al. 2022, Krapf et al. 2022, Shibata et al. 2022)

References

Ott, C., Heinmöller, E., Gaumann, A. et al. Intraepitheliale Neoplasien (PanIN) und intraduktale papillär-muzinöse Neoplasien (IPMN) des Pankreas als Vorläufer des Pankreaskarzinoms. Med Klin 102, 127–135 (2007). https://doi.org/10.1007/s00063-007-1013-8

Pittman ME, Rao R, Hruban RH. Classification, Morphology, Molecular Pathogenesis, and Outcome of Premalignant Lesions of the Pancreas. Arch Pathol Lab Med. 2017 Dec;141(12):1606-1614. doi: 10.5858/arpa.2016-0426-RA. 

Makohon-Moore AP, Matsukuma K, Zhang M, Reiter JG, Gerold JM, Jiao Y, Sikkema L, Attiyeh MA, Yachida S, Sandone C, Hruban RH, Klimstra DS, Papadopoulos N, Nowak MA, Kinzler KW, Vogelstein B, Iacobuzio-Donahue CA. Precancerous neoplastic cells can move through the pancreatic ductal system. Nature. 2018 Sep;561(7722):201-205. doi: 10.1038/s41586-018-0481-8.

Goggins M, Overbeek KA, Brand R, Syngal S, Del Chiaro M, Bartsch DK, Bassi C, Carrato A, Farrell J, Fishman EK, Fockens P, Gress TM, van Hooft JE, Hruban RH, Kastrinos F, Klein A, Lennon AM, Lucas A, Park W, Rustgi A, Simeone D, Stoffel E, Vasen HFA, Cahen DL, Canto MI, Bruno M; International Cancer of the Pancreas Screening (CAPS) consortium. Management of patients with increased risk for familial pancreatic cancer: updated recommendations from the International Cancer of the Pancreas Screening (CAPS) Consortium. Gut. 2020 Jan;69(1):7-17. doi: 10.1136/gutjnl-2019-319352. Epub 2019 Oct 31. Erratum in: Gut. 2020 Jun;69(6):e3. PMID: 31672839; PMCID: PMC7295005.

Permuth-Wey, J., & Egan, K. M. (2009). Family history is a significant risk factor for pancreatic cancer: results from a systematic review and meta-analysis. Familial cancer, 8(2), 109–117. https://doi.org/10.1007/s10689-008-9214-8

Dunne RF, Roeland EJ. The Interplay Among Pancreatic Cancer, Cachexia, Body Composition, and Diabetes. Hematol Oncol Clin North Am. 2022 Oct;36(5):897-910. doi: 10.1016/j.hoc.2022.07.001. 

Krapf SA, Lund J, Saqib AUR, Bakke HG, Rustan AC, Thoresen GH, Kase ET. Pancreatic Cancer Cell-Conditioned, Human-Derived Primary Myotubes Display Increased Leucine Turnover, Increased Lipid Accumulation, and Reduced Glucose Uptake. Metabolites. 2022 Nov 10;12(11):1095. doi: 10.3390/metabo12111095. 

Shibata C, Otsuka M, Seimiya T, Kishikawa T, Ishigaki K, Fujishiro M. Lipolysis by pancreatic cancer-derived extracellular vesicles in cancer-associated cachexia via specific integrins. Clin Transl Med. 2022 Nov;12(11):e1089. doi: 10.1002/ctm2.1089. 

We thank the reviewer for the hint and have rephrased the 5th paragraph (line 84 - 94) in the section Background as follows and added the above-mentioned references:

Since PDAC is often diagnosed at an advanced stage when patients show symptoms of major changes in metabolic processes, e.g. cachexia, proteomic and/or metabolic profiles of PDAC patients may already have been altered substantially [15]–[17]. Therefore, we aim to discover a robust subset of serum biomarkers for detection of significant lesions prior to or at an early, asymptomatic stage of cancer. To achieve this goal, we investigated individuals at risk (IARs) of FPC families with three different phenotypes: those without or with lesions detected on imaging by either magnetic resonance imaging (MRI) and/or endosonography, and those with histologically significant lesions. The latter include high-grade pancreatic intraepithelial neoplasms (PanIN) and intraductal papillary-mucinous neoplasms (IPMN) with dysplasia which are considered true precursor lesions of PDAC [12], [18]–[21]. Significant lesions, however, are rarely discovered in IARs, as they can only be confirmed histologically after the pancreatic surgery. This fact leads to small sample sizes and an unbalanced study design in the FaPaCa data.

Reviewer #1 (continued): Identified biomarkers need to be validated in another set.

Our reply:

We agree that an independent set of samples is necessary to validate the discovered biomarkers further to generalize the findings presented in our work. However, given the currently available set of samples, such a validation would be beyond the scope of this work. Hence, to validate our findings to the best possible degree at least internally, we performed cross-validation within our samples to show the effectiveness of our statistical learning models and the discovered biomarkers in predicting the left-out individuals’ statuses. In this context, we also employed the statistical method termed stability selection. Since we could detect a stable set of biomarkers with high potential to improve diagnostics, we consider our findings and the statistical analysis guideline for handling the small unbalanced sample sets important for the community.

However, we see the point to address the issue of clinical validation and stability of the selected biomarker sets and rephrased the second part of the paragraph Strengths and Limitations (line 456 - 461) as follows:

Hence, we reduced the flexibility of statistical learning methods to maintain the interpretability of the results and introduced an additional layer of stability selection to identify plausible and stable subsets of biomarkers. Furthermore, we conducted simulation studies for a thorough investigation of the validity and robustness of our findings. We consider our analyses to be an important hypothesis-generating step, to be succeeded by several subsequent clinical evaluation and validation procedures. For future research, it would be of interest to define a semi-supervised screening procedure that integrates prior clinical knowledge about protein networks, allows prioritizing variables within known clusters, and defines a subset of biomarkers that is hard-wired to be included in the model.

Reviewer #2: The submitted manuscript by Ha et al applied advance simulation methods, like adaptive lasso to analyze the proteomic data to identify the promising biomarkers for the pancreatic cancer. The authors concluded that application of these advance learning methods improve the predictive accuracy of the proteomic data and provided the examples of FGF-BP1, PCSK9, and MSLN in pancreatic cancer inception and progression. Previous studies have also implicated FGF-BP1, and MSLN in pancreatic cancer pathology. Overall, the manuscript is well written and presented to understand the application of advance statistical methods. Moreover, authors addressed all the previous concerns.

Our reply: 

Thank you very much for reviewing our paper and appreciating our results and statistical learning methods applied in our work.

7. PLOS authors have the option to publish the peer review history of their article (what does this mean?). If published, this will include your full peer review and any attached files.

Do you want your identity to be public for this peer review? For information about this choice, including consent withdrawal, please see our Privacy Policy.

Reviewer #1: No

Reviewer #2: No

---

## [Editor Report · Decision Letter 2]

28 Dec 2022

Proteomics biomarker discovery for individualized prevention of familial pancreatic cancer using statistical learning

PONE-D-22-07771R2

Dear Dr. Ha,

We’re pleased to inform you that your manuscript has been judged scientifically suitable for publication and will be formally accepted for publication once it meets all outstanding technical requirements.

Kind regards,

Surinder K. Batra

Academic Editor

PLOS ONE
---

## [Editor Report · Acceptance letter]

18 Jan 2023

PONE-D-22-07771R2 

Proteomics biomarker discovery for individualized prevention of familial pancreatic cancer using statistical learning 

Dear Dr. Ha:

I'm pleased to inform you that your manuscript has been deemed suitable for publication in PLOS ONE. Congratulations! Your manuscript is now with our production department. 

Kind regards, 

on behalf of

Prof. Surinder K. Batra 

Academic Editor

PLOS ONE